# Development of a clinical prediction score for Ebola virus disease screening at triage centers in the Democratic Republic of the Congo

Jepsy Yango[1,2]*, Antoine Oloma Tshomba[1], Papy Kwete[1], Joule Madinga[1,3], Sabue Mulangu[1,3], Placide Mbala-Kingebeni[1,3], Aquiles R. Henriquez-Trujillo[2], Bart K. M. Jacobs[2]

1 Department of Epidemiology and Global Health, Institut National de Recherche Biomédicale, Kinshasa, Democratic Republic of the Congo, 2 Department of Clinical Sciences, Institute of Tropical Medicine, Antwerp, Belgium, 3 Department of Medical Biology, University of Kinshasa, Kinshasa, Democratic Republic of the Congo

* jepsyango@gmail.com

**Data Availability Statement:** All relevant data supporting the findings of this study are included in the manuscript and its Supporting information

## Abstract

The 2018–2020 Ebola virus disease (EVD) outbreak in the Democratic Republic of the Congo (DRC) was the largest since the disease's discovery in 1976. Rapid identification and isolation of EVD patients are crucial during triage. This study aimed to develop a clinical prediction score for EVD using clinical and epidemiological predictors. We conducted a retrospective cross-sectional study using surveillance data from EVD outbreak, collected during routine clinical care at the Ebola Transit Center (ETC) in Beni, DRC, from 2018 to 2020. The Spiegelhalter and Knill-Jones method was used for score development, including potential predictors with an adjusted likelihood ratio above 2 or below 0.50. Validation was performed using a dataset previously published in PLOSOne by Tshomba et al. Among 3725 patients screened, 3698 fulfilled the inclusion criteria, with 571 (15.4%) testing positive for EVD via RT-PCR Test. Seven predictive factors were identified: asthenia, sore throat, conjunctivitis, bleeding gums, hematemesis, contact with a sick person, and contact with a traditional healer. The prediction score achieved an Area under the receiver operating characteristic (AUROC) of 0.764, with 81.4% sensitivity and 53.6% specificity at a -1 cutoff. External validation demonstrated an AUROC of 0.766, with 80.8% sensitivity and 41.4% specificity at the -1 cutoff. Our study developed a screening tool to assess the risk of suspected patients developing EVD and being admitted to ETUs for RT-PCR testing and treatment. External validation results affirmed the model's reliability and generalizability in similar settings, suggesting its potential integration into clinical practice. Given the severity and urgency of EVD as well as the risk nosocomial EVD transmission, it is essential to continuously update these models with real-time data on symptoms, disease progression, patient outcomes and validated RDT during EVD outbreaks. This approach will enhance model accuracy, enabling more precise risk assessments and more effective outbreak management.

files. These materials are available for download and review.

**Funding:** The authors received no specific funding for this work.

**Competing interests:** The authors have declared that no competing interests exist.

## Introduction

Ebola virus disease (EVD) is a severe illness with an average case fatality rate of approximately 50%, ranging from 25% to 90% depending on the Ebola species, circumstances, and responses [1]. Symptoms appear after an incubation period of 2 to 21 days. EVD manifests as a severe systemic illness characterized by fever, fatigue, myalgia, headache, pharyngitis, vomiting, diarrhea, and rash, among others [1]. Ebola virus disease (EVD) is caused by the Ebola virus (EBOV), which is now classified under the species Orthoebolavirus zairense, a member of the Filoviridae family in the genus Orthoebolavirus. Different species within this genus exist, but the most virulent and commonly associated with human outbreaks are Orthoebolavirus zairense, Orthoebolavirus sudan and Bundibugyo[2]. The virus is transmitted from wild animals to humans and then spreads via human-to-human transmission[3] The earlier the disease is diagnosed and managed, the greater the chance of recovery [4, 5].

The Ebola virus was first discovered in the Democratic Republic of Congo (DRC) in 1976, then the country has experienced several epidemics since then [5]. Between 2013 and 2016, West Africa experienced the largest and most complex outbreak in history, involving three countries: Guinea, Liberia, and Sierra Leone. A few years later, from 2018 to 2020, the DRC experienced another significant epidemic in the eastern part of the country, described as one of the largest ever recorded in the country and the second largest in the world [4, 5].

The Ministry of Health of the DRC led the outbreak response with support from the World Health Organization (WHO) and other Ministry of Health partners. The priorities of this response were focused on control measures including rapid case detection, isolation, and treatment to interrupt the chain of transmission [6]. Initially, EVD suspects were stratified in triage areas according to an EVD case definition established by WHO. Following the WHO case definition, the admission of symptomatic patients to the Ebola treatment units (ETUs) was based on a positive reverse transcription polymerase chain reaction (RT-PCR) test (GeneXpert) [4]. The response was hampered by poor infrastructure, limited access to GeneXpert machines due to security context (such as curfews and conflict zones), logistical and geographical barriers, and lack of reliable rapid diagnostic tests (RDTs) [7]. These challenges make it difficult to rapidly identify and isolate patients with presumptive EVD and to quickly provide GeneXpert results. Moreover, the WHO case definition of active EVD had a sensitivity of 81.5% and a low specificity of 35.7%. This leads to a poor capacity to discriminate EVD from other illnesses and increases the risk of nosocomial infection at the triage point [8].

This challenge was highlighted during the West African epidemics, leading to several studies aimed at predicting EVD [8–10]. For instance, Levine et al. (2015) conducted a study in Liberia using a predictive model, identifying factors such as contact with a sick person, diarrhea, loss of appetite, muscle aches, difficulty swallowing, and absence of abdominal pain, with an AUROC of 0.75. More recently, in 2022, Tshomba AO et al. conducted a study in Butembo town, DRC, using two prediction scores: the Clinical Prediction Score (CPS) based on clinical predictors, and the Extended Clinical Prediction Score (ECPS) incorporating clinical, epidemiological, and sociodemographic predictors, utilizing the Spiegelhalter method. Their study identified predictive factors including fatigue, difficulty swallowing, red eyes, gingival bleeding, hematemesis, confusion, hemoptysis, and a history of contact with an EVD case, with AUROCs of 0.71 for CPS and 0.88 for ECPS [11].

These studies highlighted the need for an optimal clinical algorithm to complement existing Ebola case definitions in future outbreaks [8, 9, 11]. Additionally, they emphasized the importance of developing and integrating rapid diagnostic tests (RDTs) into predictive models due to the inherent limitations of current predictive models [12]. To improve decision-making processes, especially at the triage point during EVD outbreaks, our study aims to develop a

clinical prediction score based on an initial risk stratification at the transit center. This score will assist healthcare workers (HCWs) in determining, in the absence of rapid testing, which patients should immediately be isolated for further investigation during EVD outbreaks.

## Methods

### Study design

This was a retrospective cross-sectional study using surveillance data of the 2018–2020 EVD outbreak extracted from the 2018–2020 Ebola surveillance database.

### Study setting

The study was conducted at the Beni Transit Center, located near the Ebola Treatment Center (ETC) in Beni town. This Transit Center was established by Médecins Sans Frontières (MSF) to enhance care capacities and receive all patients with presumptive EVD pending laboratory confirmation [13].

Depending on their clinical condition, patients received first aid while awaiting EVD RT-PCR test (GeneXpert) results. Patients with a positive EVD test were referred to the ETU, while those without EVD were either referred to other health facilities or discharged home [13].

### Study population

**Inclusion and exclusion criteria.** The study included all individuals who were referred to the Beni transit center from August 2018 to February 2020, either by healthcare facilities for suspected Ebola virus disease, as per the WHO case definition, as well as those identified through active community surveillance and investigations, or those who voluntarily arrived at the transit center. Individuals who died on arrival, and for whom no symptom data were collected, were excluded from the study.

### Data collection procedure

Patients or their family members were interviewed by HCWs to collect personal information, symptoms, duration of illness, and epidemiological risk factors, such as attendance at funerals, handling of corpses, and contact with sick patients or family members.

Interviews were conducted to describe the symptoms according to the WHO case definition. National and international HCWs in transit centers were trained to conduct interviews and complete standardized paper forms. All data were recorded in the 2018–2020 Ebola Surveillance Database.

Cepheid's Ebola GeneXpert semi-automated closed system reverse transcription-quantitative polymerase chain reaction (RT-qPCR) platform was used for the diagnosis of EVD during this outbreak following the WHO instructions as the reference standard for the diagnosis of EVD at the point of care. This technique offers a lower risk of contamination, high sensitivity ($>$ 99%) and specificity ($>$95%) and is fast and easy to use with minimal technical knowledge and a short turnaround time ($<$ 2 h). [14–17].

### Statistical analysis

To develop the scoring system, we used the Spiegelhalter and Knill-Jones method adapted by Berkley et al. [18–20]

This approach combines principles from Bayesian inference (independent Bayes method) and logistic regression to construct a scoring system [18–20]. Relevant clinical predictors are

selected, and their likelihood ratios are calculated to determine their association with the disease. Logistic regression refines these predictions, adjusting for multiple predictors. Scores are assigned based on the natural logarithm of the adjusted likelihood ratios, which are summed to generate a total score for each patient [11, 21].

- Positive Likelihood Ratio (LR+) indicates how much more likely a positive test result is to occur in someone with the disease compared to someone without the disease. An LR+ greater than 1 suggests that the symptom is associated with the disease.

$$\text{LR+} = \frac{Sensitivity}{1 - Specificity}$$

- Negative Likelihood Ratio (LR-) indicates how much less likely a negative test result is to occur in someone with the disease compared to someone without the disease. An LR- less than 1 suggests that the absence of the symptom is associated with not having the disease.

$$\text{LR-} = \frac{1 - Sensitivity}{Specificity}$$

Considering GeneXpert as the reference test, we calculated the sensitivity, specificity, positive likelihood ratio (LR+) and negative likelihood ratio (LR-) for each clinical sign or symptom associated with EVD [11, 17, 21].

All clinical and epidemiological predictors with crude positive likelihood ratios > 2 and negative likelihood ratios < 0.5 were included, and logistic regression was performed on likelihood ratios [18, 20]. The score was generated based on the natural logarithm of the adjusted likelihood ratios for each predictor (with the value 0 assigned to missing data), rounding these results to the nearest integer [11, 21] and adding the values of all the patient-presented predictors.

$$Score = Round\left(ln\left(Adjusted\ LR\right]\right)$$

The discriminatory capacity or performance of this score was evaluated using a receiver operating characteristic (ROC] curve [11, 21], and the diagnostic accuracy of the scoring system was assessed by calculating the sensitivity and specificity of the score at different cut-off values [11, 20, 21]. All analyses were performed using R (version 4.2.3; R Foundation for Statistical Computing, Vienna, Austria) in R-Studio (version 2023.03.1+446, RStudio, PBC, Boston, MA, USA).

### External validation

Following the development of the scoring model, we conducted external validation by applying our predictors to the population described by Tshomba AO et al. using their database available on PLOS One. The objective of this validation process was to evaluate the model's performance across diverse datasets and populations, ensuring its reliability and effectiveness for practical use.

The dataset by Tshomba Ao et al comprises 10,432 subjects, including 651 confirmed cases of EVD, indicating a prevalence of 6.2%. These data were collected during the tenth EVD epidemic in Butembo, DRC. The dataset was utilized to evaluate EVD predictors against GeneXpert results and to develop Clinical Prediction Scores (CPS) and Extended Clinical Prediction Scores (ECPS) using the Spiegelhalter-Knill-Jones method [11]. Key predictors of EVD include

fatigue, difficulty swallowing, red eyes, gingival bleeding, hematemesis, confusion, hemoptysis, and a history of contact with EVD cases. The ECPS demonstrated a significantly higher AUROC than the CPS [11].

## Ethics

The present study was granted ethical approval by the Ethics Committee of the Democratic Republic of the Congo (ESP/CE/132/2022), as well as by the Institutional Review Board of the Institute of Tropical Medicine in Belgium (IRB/RR/AC/079).

Due to the study's retrospective design and use of pre-existing data, participants did not require informed consent. All personal information from Ebola Surveillance Database for the study period was anonymized to ensure confidentiality and to comply with data protection regulations.

## Results

Between 2018 and 2020, the surveillance system recorded data from 3,725 patients suspected of having Ebola Virus Disease (EVD) in Beni, North Kivu province. Of these, 3,698 participants met the study's inclusion criteria. Among the participants, 571 (15.4%) tested GeneXpert positive for EVD, while 3,127 (84.6%) tested negative (Fig 1).

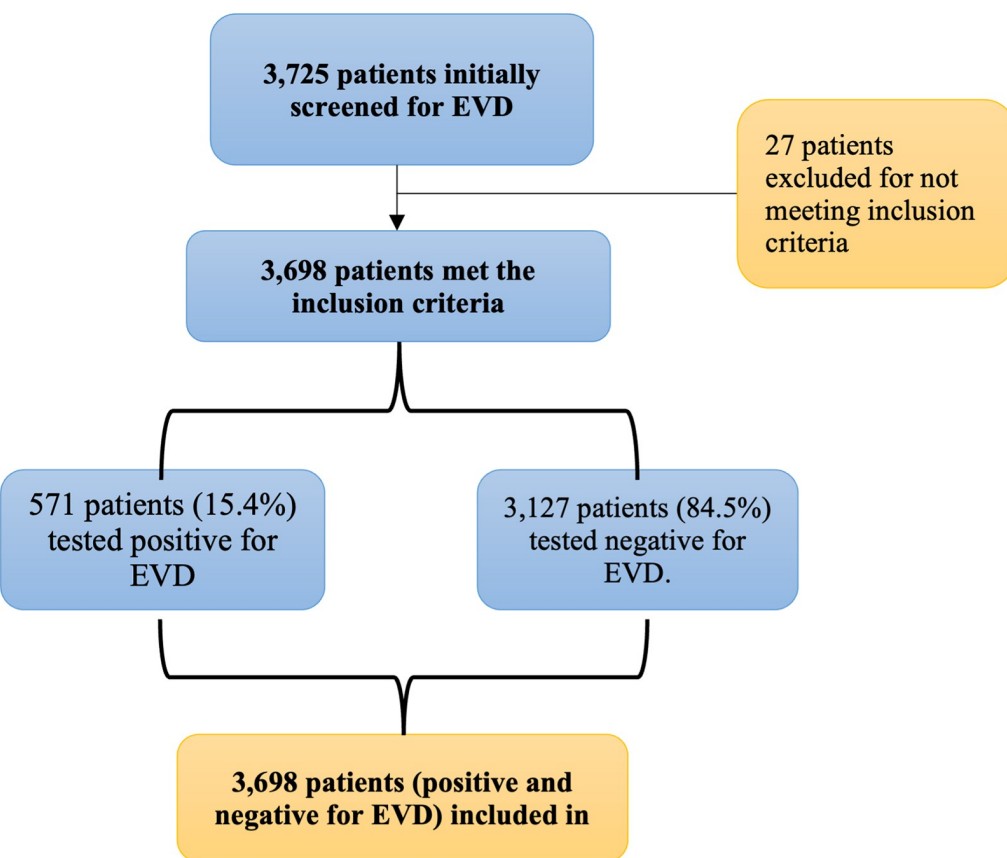

**Fig 1. The flow diagram illustrates the selection process of patients for the clinical prediction score (CPS) study on Ebola virus disease (EVD).**

**Table 1. Sociodemographic characteristics of participants in relation to the outcome.**

| Characteristic | Total | EVD positive | EVD negative | P-value[1] |
|---|---|---|---|---|
| | *n = 3698* | *n = 571* | *n = 3127* | |
| **Age, median (IQR)** | 22 (10, 33) | 26 (17, 38) | 21 (9, 32) | <0.001 |
| *Unknown* | 8 | 0 | 8 | |
| **Sex, n (%)** | | | | <0.001 |
| *Female* | 1882 (50.8%) | 333 (58.3%) | 1549 (49.5%) | |
| *Male* | 1816 (49.1%) | 238 (41.6%) | 1578 (50.4%) | |
| **Profession, n (%)** | | | | <0.001 |
| *Farmer* | 719 (19.4%) | 152 (26.6%) | 567 (18.1%) | |
| *HCWs* | 111 (3.0%) | 32 (5.6%) | 79 (2.5%) | |
| *Hunter* | 136 (3.7%) | 25 (4.4%) | 111 (3.5%) | |
| *No Profession* | 56 (1.5%) | 15 (2.6%) | 41 (1.3%) | |
| *Other* | 1148 (31.0%) | 185 (32.3%) | 963 (30.7%) | |
| *Student* | 1528 (41.3%) | 162 (28.3%) | 1366 (43.6%) | |

[1] Wilcoxon rank sum test (age) and Pearson's chi-squared test (sex, profession)

The sociodemographic characteristics of the participants are summarized in Table 1. The median age of the patients with EVD was 26 years (IQR: 17–38). Among EVD-positive patients, 58.3% were female compared to 49.5% of EVD-negative patients. Regarding professions, the majority of EVD-positive patients were from other occupations (32.3%), followed by students (28.3%) and farmers (26.6%). In contrast, among EVD-negative patients, students (43.6%) were the majority, followed by other occupations (30.7%).

The most common symptoms among EVD-positive participants were fever (53.2%), asthenia (79.9%), and loss of appetite (69.9%). Table 2 shows the diagnostic accuracy measures for various clinical and epidemiological predictors. Nine predictors met the inclusion criteria based on their likelihood ratios: asthenia (LR-: 0.50), sore throat (LR+: 2.37), conjunctivitis (LR+: 3.88), bleeding gums (LR+: 4.11), hematemesis (LR+: 2.02), contact with a sick person (LR+: 3.69, LR-: 0.46), contact with a traditional healer (LR+: 2.27), attending a funeral (LR+: 3.47), and purpura (LR+: 2.74).

The clinical prediction score (CPS) was developed using the Spiegelhalter and Knill-Jones method, as adapted by Berkley et al. Sensitivity, specificity, positive likelihood ratio (LR+), and negative likelihood ratio (LR-) for each clinical sign or symptom associated with EVD are detailed in Table 2. The scoring system was further refined using logistic regression based on adjusted likelihood ratios, resulting in the final scores (Table 3).

ROC curve analysis demonstrated an Area Under the Receiver Operating Characteristic (AUROC) of 0.764 in the training data (Fig 2). This AUROC value indicates that the CPS has a satisfactory ability to differentiate between patients with and without EVD in the training dataset. When applied to the external validation dataset from Tshomba AO et al., the AUROC was 0.766 (Fig 2), demonstrating similar performance. This consistency between the training and external validation datasets indicates that the CPS is a reliable predictor of EVD in similar population.

Table 4 provides a detailed analysis of the diagnostic performance of the Clinical Prediction Score (CPS) at different cut-off points for predicting EVD. At a cut-off point of -1, the sensitivity was 81.4% and the specificity was 53.6%, indicating that the CPS is quite sensitive but less specific at this cut-off point. In addition, after external validation, the performance of the CPS showed a sensitivity of 80.8% and a specificity of 41.4% at the same cut-off point, reflecting a

**Table 2. Diagnostic accuracy of clinical and epidemiological predictors of EVD.**

| Predictors | Sensitivity | Specificity | LR+ | LR- |
|---|---|---|---|---|
| Fever | 53.24% | 50.53% | 1.08 | 0.93 |
| Nausea | 50.44% | 51.17% | 1.03 | 0.97 |
| Diarrhea | 33.98% | 67.25% | 1.04 | 0.98 |
| Asthenia | 79.96% | 40.10% | 1.31 | 0.50 |
| Loss of appetite | 69.90% | 34.67% | 1.07 | 0.87 |
| Abdominal pain | 42.93% | 43.97% | 0.76 | 1.30 |
| Chest pain | 20.14% | 83.18% | 1.20 | 0.96 |
| Bone and muscle pain | 35.20% | 72.24% | 1.27 | 0.90 |
| Joint pain | 43.78% | 66.17% | 1.29 | 0.85 |
| Headache | 52.59% | 42.95% | 0.91 | 1.11 |
| Cough | 16.46% | 71.60% | 0.58 | 1.17 |
| Breathlessness | 9.16% | 86.92% | 0.69 | 1.05 |
| Swallowing problem | 16.46% | 91.17% | 1.87 | 0.92 |
| Sore throat | 15.41% | 93.51% | 2.37 | 0.90 |
| Jaundice | 2.45% | 98.17% | 0.78 | 1.01 |
| Conjunctivitis | 6.83% | 98.24% | 3.88 | 0.95 |
| Rash | 2.80% | 97.86% | 1.31 | 0.99 |
| Hiccups | 4.38% | 97.54% | 1.78 | 0.98 |
| Eyes pain | 2.63% | 98.15% | 1.42 | 0.99 |
| Coma | 1.23% | 96.61% | 0.36 | 1.02 |
| Confusion | 1.75% | 98.13% | 0.81 | 1.00 |
| Bleeding gum | 1.60% | 99.62% | 4.11 | 0.99 |
| Epistaxis | 3.33% | 97.83% | 1.29 | 0.99 |
| Melena | 3.63% | 94.92% | 0.65 | 1.02 |
| Hematemesis | 4.10% | 98.18% | 2.02 | 0.98 |
| 'Vomito negro' or black vomit | 0.40% | 99.58% | 0.64 | 1.00 |
| Hemoptysis | 0.20% | 99.36% | 0.24 | 1.01 |
| Bleeding vagina | 1.15% | 95.71% | 0.25 | 1.03 |
| Purpura | 0.41% | 99.97% | 2.74 | 1.00 |
| Bleeding urine | 0.41% | 99.17% | 0.38 | 1.01 |
| Contact with a sick person | 61.47% | 83.34% | 3.69 | 0.46 |
| Funeral attendance | 28.55% | 91.78% | 3.47 | 0.78 |
| Contact with a sick traveler | 14.11% | 89.22% | 1.30 | 0.96 |
| Contact with a traditional healer | 4.10% | 98.24% | 2.27 | 0.97 |

LR+: Positive likelihood ratio, LR−: Negative likelihood ratio

consistent sensitivity but a decrease in specificity. This table shows the performance of the CPS at different cut-off points and helps to understand the trade-offs between sensitivity and specificity in the prediction of EVD.

## Discussion

The clinical prediction score developed in this study identifies seven factors that effectively predict a patient's risk of developing Ebola Virus Disease (EVD): sore throat, asthenia, conjunctivitis, bleeding gums, hematemesis, contact with a sick person, and contact with a traditional healer. The score, combined with the external validation, demonstrated satisfactory

**Table 3. Score per predictor assigned after logistic regression analysis based on the Spiegelhalter Knill-Jones method.**

| Predictors | LR+ | LR- | aLR+ | aLR- | Score + | Score - |
|---|---|---|---|---|---|---|
| Asthenia[1] | 1.31 | 0.50 | 1.17 | 0.67 | 0 | -1 |
| Sore throat | 2.37 | 0.90 | 2.19 | 0.91 | 1 | 0 |
| Conjunctivitis | 3.88 | 0.95 | 2.01 | 0.97 | 1 | 0 |
| Bleeding gum | 4.11 | 0.99 | 4.78 | 0.98 | 2 | 0 |
| Hematemesis | 2.02 | 0.98 | 2.60 | 0.97 | 1 | 0 |
| Contact with a sick person | 3.69 | 0.46 | 3.83 | 0.45 | 1 | -1 |
| Contact with a traditional healer | 2.27 | 0.98 | 2.05 | 0.98 | 1 | 0 |

LR+, positive likelihood ratio; LR−, negative likelihood ratio; aLR+; Adjusted positive likelihood ratio; aLR−: Adjusted negative likelihood ratio

[1] Asthenia was assigned a score of -1 when absent despite not reaching the theoretical cut-off in the multivariate step.

performance with an Area Under the Receiver Operating Characteristic curve (AUROC) of 0.764, indicating its potential usefulness in distinguishing EVD from other illnesses. (Fig 2).

Our study's findings indicate that the predictors exhibited a sufficient degree of sensitivity and specificity, which suggests their potential utility in identifying suspected Ebola patients at the triage point. At the critical cut-off point of -1 or higher, sensitivity stood at 81.4%, while specificity reached 53.6%. This shows a higher accuracy compared to the CPS proposed by Tshomba et al., particularly in terms of specificity (42.3% vs. 53.6%). Following validation using the Tshomba et al. dataset, our CPS reached an Area Under the Curve (AUROC) of 0.766, which is similar to the performance in the training dataset. This AUROC indicates an apparent reliability of our CPS, suggesting its applicability in diverse settings beyond the immediate scope of the study. However, further validation in even more varied contexts is warranted to establish its generalizability as the validation dataset was collected during the same Ebola wave and in the same country.

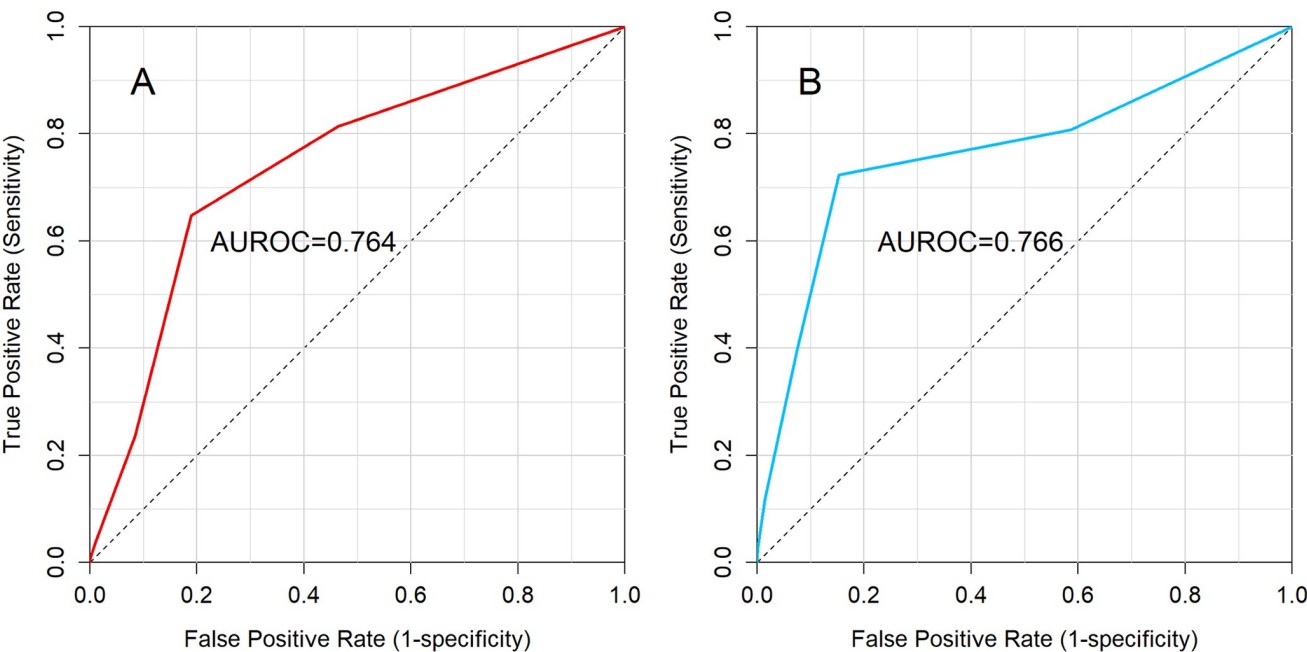

**Fig 2. CPS ROC (panel A, left) and Validation ROC (panel B, right).**

**Table 4. Diagnostic accuracy of CPS and external validation at various score cut-off points.**

| | CPS | | | External validation | | |
|---|---|---|---|---|---|---|
| Scores | Sensitivity | Specificity | AUROC | Sensitivity | Specificity | AUROC |
| < -2 vs. > = -2 | 100 | 0 | **0.764** | 100.0 | 0.0 | **0.766** |
| < -1 vs. > = -1 | 81.4 | 53.6 | | 80.8 | 41.4 | |
| < 0 vs. > = 0 | 64.8 | 81.0 | | 72.4 | 84.7 | |
| < 1 vs. > = 1 | 23.6 | 91.6 | | 39.8 | 92.5 | |
| < 2 vs. > = 2 | 4.4 | 98.8 | | 11.8 | 98.5 | |
| < 3 vs. > = 3 | 1.1 | 99.9 | | 2.8 | 99.9 | |
| **< 4 vs. > = 4** | 0.4 | 100.0 | | 0.9 | 100.0 | |

Studies conducted in West Africa showed similar results [9, 10]. Levine et al. 2015 [10], identified sick contact, diarrhea, loss of appetite, muscle pain, and swallowing problems with an AUROC of 0.75. Loubet et al. 2016 [9] found fever (>38°C), fatigue, anorexia, and gastrointestinal problems as predictors with an AUROC of 0.80. Our clinical prediction score revealed predictors similar to those identified by Tshomba et al. in 2022 and demonstrated satisfactory performance (AUROC: 0.764). In contrast, Tshomba et al.'s CPS had an AUROC of 0.71. However, our CPS demonstrated a higher AUROC when applied to Tshomba's dataset for validation, despite Tshomba using it as a training dataset. This observation is remarkable, especially considering that both studies employed similar methodologies. [11]

It is important to acknowledge that admitting a patient unlikely to have EVD to an Ebola treatment center poses the risk of nosocomial infection, while also reducing the timely identification and appropriate treatment of true EVD cases [11]. While awaiting confirmation from molecular tests such as GeneXpert, clinicians must judiciously decide which patients require admission for conclusive testing and treatment. At this critical juncture, clinicians must exercise their best judgment to make informed decisions that aim to limit and break the chain of transmission directly at the triage site. In the absence of validated rapid diagnostic tests, clinical signs, symptoms, and epidemiological information serve as intermediate criteria for identifying EVD suspects and managing patients. However, the similarity in symptomatology between EVD and other infectious diseases complicates decision-making at the triage point, given the low specificity of early signs and EVD-specific signs appearing only in the terminal phase [11].

In daily practice, this score could be utilized as follows in initial risk stratification at the transit center to determine immediate isolation, and optimally utilize the available RT-PCR testing capacity:

- Patients with a score below -1 (score -2) represent a low-risk group; only in absence of available RT-PCR testing, they could potentially be sent home, however, due to the lethality of the disease and the probability of disease of approximately 6% in this group, they should still be closely followed up within 21 days and tested as soon as the capacity allows for it.

- Patients with a score of -1 or 0 have an approximate prevalence of 10% compared to the included population, which had a prevalence of 15.4%; therefore, they represent a medium-risk group. They should be isolated and tested as soon as possible and followed up for 21 days.

- A positive cut-off of +1 or higher represents a high-risk group with a high probability of the disease and requires immediate investigation or isolation, and a cut-off of +2 indicates a high probability of the disease and a priority for confirmatory RT-PCR testing.

A practical algorithm would be:

- In the event of contact with a sick person: score of +1, then check for other predictors; if at least one other predictor is present, the patient is considered high-risk because the score is at least +1. If no other predictors are present, the patient is considered to be at medium risk (score = 0).

- If there is no contact with a sick person, then check: if none of the other predictors mentioned above are present, the patient is considered low risk, and the score is -2.

- If there is no contact with a sick person but at least one and less than three predictors (including bleeding gums counting double), the patient is considered to be at medium risk, as the score is either 0 or -1. If there is no contact with a sick person but at least three predictors (bleeding gums counting double), the patient is considered high risk, and the score is at least +1. (Fig 3)

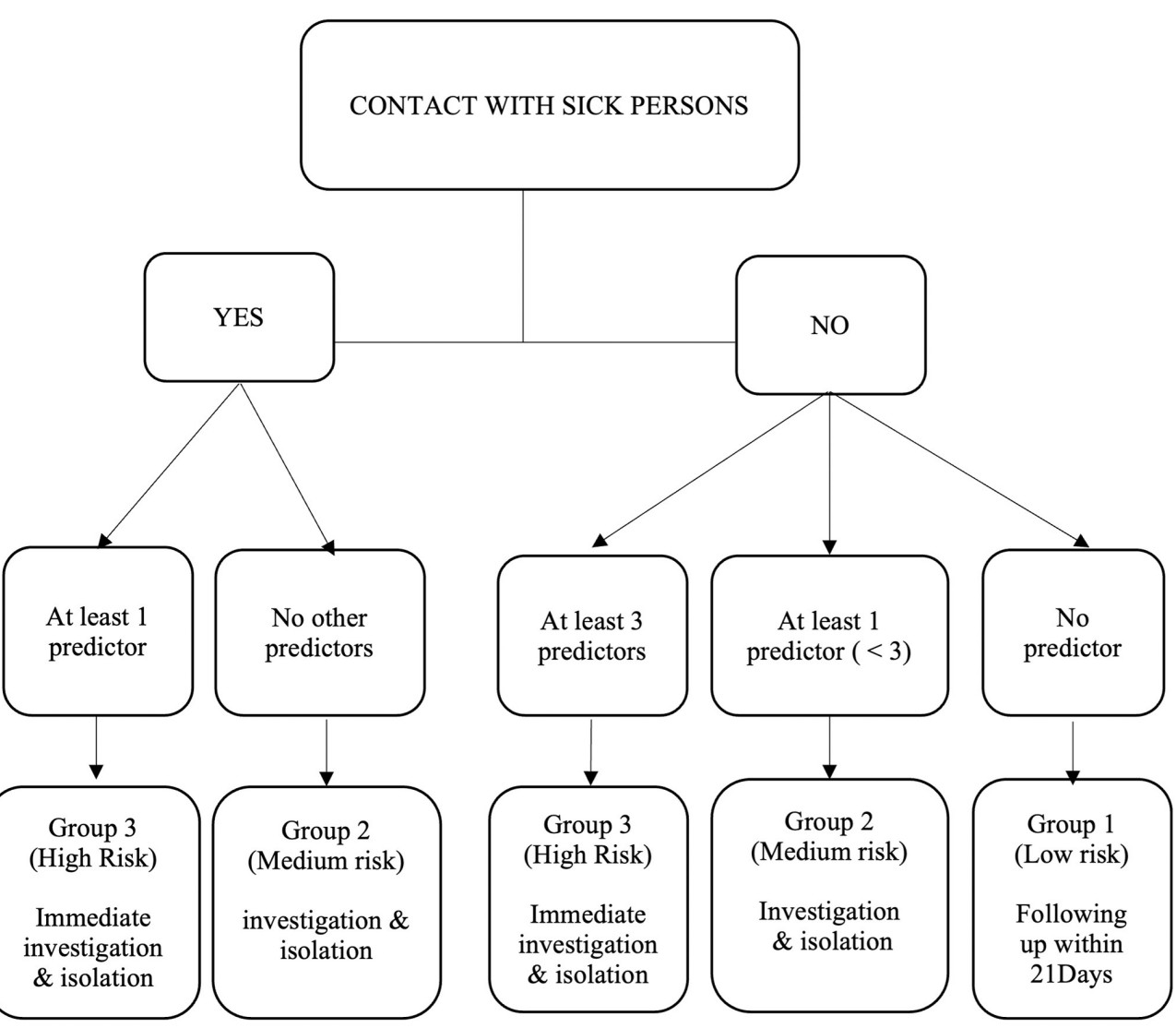

**Fig 3. Decision tree for using the CPS.** The other predictors are asthenia, sore throat, conjunctivitis, bleeding gum, hematemesis and contact with a traditional healer, with bleeding gum counting double.

We can apply this algorithm to all suspected cases. Patients with high or medium scores will be isolated, while others will not be isolated but should undergo follow-up within 21 days.

Tshomba et al. highlighted in their study that test sensitivity can appear better in the suspected population with less severe EVD suspects [11]. In our study, we noted a prevalence of 15.4%, with data collected at the peak of the epidemic from suspects already showing signs of severity, which influenced our results in terms of sensitivity and specificity.

Therefore, this score can only be applied in epidemic contexts, environments with limited resources, and poor infrastructures, where RDT or RT-PCR tests are lacking [7, 12]. It can be used as a triage tool, to help reduce delays in isolation and patient management, and to reduce the number of unnecessary hospitalizations. However, clinical judgment is important, and this score is designed to help clinicians make better decisions.

The scoring system developed in this study may help to evaluate the risk of Ebola virus disease (EVD) in individuals presenting at triage centers with presumptive EVD symptoms. Its simplicity and standardization render it particularly valuable in resource-constrained environments and amidst high-incidence Ebola epidemics. Clinicians can leverage this tool for prompt case triaging, informed decision-making, and effective case management at the triage point. Its utility extends to reducing the decision-making time for health workers during patient management, thereby contributing to timely interventions. Additionally, the tool holds promise in preventing nosocomial infections by facilitating the rapid identification and isolation of potential EVD cases. Moreover, in scenarios where the number of suspected EVD cases exceeds the standardized capacity of healthcare facilities, the CPS proves to be an essential tool for triage. It allows HCWs to quickly assess and prioritize patients based on their risk level, ensuring that those who need immediate attention are isolated and tested promptly. While this tool shows promising attributes, external validation is essential before deploying it in diverse contexts. In settings like the DRC, characterized by the same EBOV variant and high prevalence rates, and with similar data collection methods, the tool is expected to perform equally well, as validated. However, in significantly different contexts, such as West Africa, where variant vary, vaccination rates are higher, and prevalence is lower, performance may vary.

The CPS can be effectively applied at different stages of an EVD outbreak, ensuring continuous and effective disease management. At the onset of an outbreak, the score helps rapidly identify and isolate high-risk individuals, preventing the spread of the disease. Early in an outbreak, with a high volume of suspected cases, the score aids in managing the surge by prioritizing those most likely to have EVD. Towards the end of an outbreak, when cases decrease, the score remains valuable for accurately identifying the few remaining cases, preventing a resurgence. By being adaptable to different stages, the CPS provides a flexible and structured approach to managing EVD outbreaks, enhancing the ability of HCWs to respond promptly and effectively throughout the epidemic.

This study has the following limitations. The use of a retrospective database exposed selection bias and the database used for this analysis was collected only at admission and contained many community-based cases from surveillance, for which clinical and epidemiological data were missing. Additionally, the study was conducted in a high-prevalence population with patients in severe stages of disease during an epidemic period, which is why we used late-stage symptoms as predictors, and the study missed most clinically asymptomatic or intermediate cases. Although the performance of our scores is good, their use in inter-epidemic periods will require further evaluation.

Furthermore, Clinical prediction tools used for triage decision-making present inherent risks. An overly sensitive tool, prioritizing the inclusion of all potential cases, may lead to the admission of numerous suspects without EVD, increasing the risk of nosocomial infection. Conversely, an overly specific tool, prioritizing the exclusion of false positives, may

prematurely release infected individuals back into the community with a presumed negative EVD diagnosis, with potentially new chains of transmission.

Given the high stakes of EVD, nucleic acid amplification tests (NAAT), other molecular methods, or rapid antigen tests, remain indispensable for effective outbreak management. Future research should prioritize the integration of affordable quality tests for EVD into triage approaches and prediction models [7, 12].

## Conclusion

This clinical prediction tool can enhance the triage process by identifying suspected cases with a high likelihood of developing EVD, allowing for rapid therapeutic decision-making. It helps reduce the rate of false negatives in the community and prioritizes true Ebola patients for access to RT-PCR testing and treatment at Ebola Treatment Units (ETUs). The tool is particularly suitable for implementation in low-resource settings with limited infrastructure and a lack of RDTs.

Our study developed a clinical prediction screening tool to assess the risk of suspected patients at the triage point developing EVD and being admitted to ETUs for definitive RT-PCR testing and treatment. To optimize future research, integrating RDTs into triage systems and coupling them with standardized symptom and clinical data collection during EVD outbreaks is essential. This approach not only facilitates the development of dynamic clinical prediction scores that adapt in real-time to new data but also revolutionizes outbreak management. Continuously updating data on symptoms, disease progression, patient outcomes and usage of validated RDT will refine the accuracy of these tools, thereby improving risk assessments, triage decisions, and resource allocation for more effective outbreak control.

## Supporting information

**S1 Data. EVD surveillance dataset.**
(XLS)

## Acknowledgments

The authors would like to thank Doctor. Tom Decroo for his insights and contributions to discussions of the work presented here. We are also grateful to Professor Lutgarde Lynen and Doctor Maria Zolfo for their invaluable guidance and support.

## Author Contributions

**Conceptualization:** Jepsy Yango.

**Data curation:** Jepsy Yango, Antoine Oloma Tshomba, Joule Madinga, Sabue Mulangu, Placide Mbala-Kingebeni, Bart K. M. Jacobs.

**Formal analysis:** Jepsy Yango, Papy Kwete, Placide Mbala-Kingebeni, Bart K. M. Jacobs.

**Funding acquisition:** Sabue Mulangu.

**Investigation:** Jepsy Yango.

**Methodology:** Jepsy Yango, Aquiles R. Henriquez-Trujillo, Bart K. M. Jacobs.

**Project administration:** Jepsy Yango, Sabue Mulangu, Placide Mbala-Kingebeni.

**Resources:** Jepsy Yango.

**Software:** Jepsy Yango, Papy Kwete, Bart K. M. Jacobs.

**Supervision:** Aquiles R. Henriquez-Trujillo, Bart K. M. Jacobs.

**Validation:** Jepsy Yango, Antoine Oloma Tshomba, Sabue Mulangu, Placide Mbala-Kinge-beni, Aquiles R. Henriquez-Trujillo, Bart K. M. Jacobs.

**Visualization:** Jepsy Yango.

**Writing – original draft:** Jepsy Yango, Antoine Oloma Tshomba, Aquiles R. Henriquez-Truji-llo, Bart K. M. Jacobs.

**Writing – review & editing:** Jepsy Yango, Antoine Oloma Tshomba, Joule Madinga, Sabue Mulangu, Placide Mbala-Kingebeni, Aquiles R. Henriquez-Trujillo, Bart K. M. Jacobs.

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
