## [Decision Letter · Decision Letter 0]

10 May 2024

PGPH-D-24-00337

Development of a clinical prediction score for Ebola virus disease screening at triage centers in the Democratic Republic of the Congo

Dear Dr. Yango wa yango,

Thank you for submitting your manuscript to PLOS Global Public Health. After careful consideration, we feel that it has merit but does not fully meet PLOS Global Public Health’s publication criteria as it currently stands. Therefore, we invite you to submit a revised version of the manuscript that addresses the points raised during the review process.

We look forward to receiving your revised manuscript.

Kind regards,

Maria del Pilar Fernandez

Academic Editor

Journal Requirements:

Additional Editor Comments (if provided):

Reviewers' comments:

Reviewer's Responses to Questions

**Comments to the Author**

1. Does this manuscript meet PLOS Global Public Health’s publication criteria? Is the manuscript technically sound, and do the data support the conclusions? The manuscript must describe methodologically and ethically rigorous research with conclusions that are appropriately drawn based on the data presented.

Reviewer #1: Yes

Reviewer #2: Yes

2. Has the statistical analysis been performed appropriately and rigorously?

Reviewer #1: Yes

Reviewer #2: No

3. Have the authors made all data underlying the findings in their manuscript fully available (please refer to the Data Availability Statement at the start of the manuscript PDF file)?

Reviewer #1: Yes

Reviewer #2: Yes

4. Is the manuscript presented in an intelligible fashion and written in standard English?

Reviewer #1: Yes

Reviewer #2: Yes

5. Review Comments to the Author

Reviewer #1: The authors make minor errors in referring to the taxonomy of Ebola virus. For example, at line 71 they state, ‘Different species of Ebola virus exist, with Zaire ebolavirus […]’. This is incorrect. Ebola virus (EBOV) is the sole member of the species Orthoebolavirus zairense. Other species within the genus Orthoebolavirus do indeed exist (and, confusingly, these are properly referred to as ‘ebolaviruses’ – lower case and not italicized), but they are distinct viruses, (eg Sudan virus (SUDV), member of species Orthoebolavirus sudanense and the etiological agent of Sudan virus disease (SVD)). At line 333, the authors use the term ‘Ebola strain’; this too is incorrect. It should say ‘Ebola variant’, as that correctly references the different Ebola viruses that have spilled over in DRC, West Africa, etc. Regarding correct Ebola virus nomenclature, taxonomy etc, I refer the authors to the ICTV website (https://ictv.global/report/chapter/filoviridae/filoviridae/orthoebolavirus) and a very helpful article by Kuhn et al (https://www.ncbi.nlm.nih.gov/pmc/articles/PMC6637750/). While these are technical points, it is beneficial to readers and to the filoviral literature as a whole when precise, correct terminology is used.

The authors repeatedly state that the EVD epidemic referenced in their study was the 10th such outbreak in DRC (eg, lines 78, 116-117, etc). They mistakenly include (as many others have done, including MSF) the 2012 outbreak of Bundibugyo virus (BDBV), member of species Orthoebolavirus bundibugyoense and etiological agent of Bundibugyo virus disease (BVD). While certainly a very similar virus that causes similar disease to EBOV in humans, it is not the same and the conclusions of this study should not necessarily be extrapolated to BDBV (or any of the other ebolaviruses). I think it would be best to avoid continually referring to the epidemic at hand with any reference to its number (9th, 10th, whatever), to avoid confusion. Again, see the taxonomy discussion above.

Lines 84-85 ‘Following the WHO case definition, symptomatic patients were admitted to the Ebola 85 treatment unit (ETU) and subjected to a PCR test (GeneXpert) for the diagnosis of EVD’. The chronology here seems incorrected. Actual admission to the ETU should be dependent upon a positive PCR. A similar statement is present at lines 355-357. During the west Africa epidemic, most ETUs had triage areas where patients were stratified for PCR testing before being definitively admitted to the ETU following a positive test. Perhaps it worked differently in Beni, but the authors should clarify the triage and admission process since this is integral to understanding the utility of their tool and the paper as a whole.

The authors should add further discussion regarding the risks inherent to using a tool such as this for triage decision making. A tool that is too sensitive will suggest that many patients without EVD be admitted and increase the risk of nosocomial infection by exposure to those who are truly infected within the ETU. A tool that is too specific will inevitably release truly infected individuals back to the community with a presumptive negative EVD diagnosis, and thus risk starting new chains of unrecognized transmission outside of the ETU. Thus, the stakes are so high with EVD that NAAT, rapid antigen testing, or other molecular methods are the only way to effectively manage an outbreak. The authors do generally acknowledge this, but should more specifically and definitively do so. Also, they mention the need for RDTs, but make no reference to the RDTs that are currently available (eg, OraQuick). Some explanation for this should be provided.

Reviewer #2: The manuscript is good in terms of idea and originality. Statistics are poorly supplied and explained. The manuscript suffers from 1) lack of valuable details to shed light on it, 2) unclear explanations, 3) poor english grammar, 4) strong attachment to a previous work that has unsettled the main ideas of the ongoing manuscript.

The grammar should be improved throughout the manuscript to make it readable.

Check with the Editor the appropriate checklist to be used here!

Introduction

1) Use the ongoing classification for Filoviridae. https://pubmed.ncbi.nlm.nih.gov/37537381/

2) Lines 75-76 are not well reflected as Sudan outbreak is overseen.

3) Lines 76-77: West African outbreak started in 2013. Correct.

4) Lines 81-82: the sentence could be more precise for example the MoH of the DRC…

5) Line 85: correct RT-PCR instead of PCR throughout the manuscript. If using abbreviations, first write them in full at the first citation then abbreviate for the rest…

6) Lines 86-92: Assuming that the response was hampered by poor access to the GX is not correct. On the other hand, the GX reduced the turn-around time, improved the diagnostic, and was the key tool in the success of > 200.000 samples tested throughout that outbreak. https://pubmed.ncbi.nlm.nih.gov/36573642/.

7) Lines 87-89 should be revised in line with the new statement authors will produce.

8) Lines 90-92: better to say discriminate rather than differentiate. What means the term “other conditions”? Rephrase.

9) Lines 93-103 should be rephrased to clearly show the junction authors want to do with the study objective.

10) Sentence containing the objective is very long and difficult to read: “EVD outbreaks” and “at the triage point » are stated twice.

11) Lines 111-112 : I do not see its usefulness there.

Methods

Write the methods as per the journal’s requirements.

1) Methods, not method.

2) Study settings should be rewritten to meet the Editor’s criteria. Not well written.

3) Inclusion and exclusion criteria: you only considered data from one transit center in Beni whereas there were others in the regions…Is it a personal choice or a lack of data to exploit?

4) EVD suspect-cases could also come from active surveillance in the community, investigation.

5) Lines 133-134: correct the structure of the sentence.

6) Lines 165-166: Add the brand/company, City, country for R.

7) Lines 168-170: drop one “the Tshomba AO et al” and rephrase.

8) Lines 170: The objective “was to evaluate” instead of “is to evaluate”.

9) Can authors clearly state what positive LR and negative LR really mean in the interpretation as it is not clearly defined.

10) Lines 183-184: specify comité d’éthique de l’Ecole de Santé publique de Kinshasa in french or english.

Results

1) Results should be presented as one body with reference to different objects (tables and figures) following a logical pathway. Here it is just a superposition of figures without clear logical course.

2) Add a flow diagram if possible

3) Lines 190-192 should be rephrased to allow easy reading.

4) Decimals should be consistently written as per the journal’s requirements: use comma or point, not both in the same manuscript. Lines 196-197: the sentence looks incomplete.

5) Should tables, figures go along with the manuscript or presented in separated documents?

6) Figure one could be consistently commented instead of a brief description.

7) Table 4 is poorly commented, not reflecting the content and the interpretation we would expect for such a data.

Discussion

The discussion section can be presented as one body text. Can drop most subheadings in the section. The limitations and strengths can be read without subheadings.

1) lines 251-252: It is not clear to me whether the score clearly helped to discriminate EVD from other illness. The performance the ROC is purely theoretical, and I did not see how the discrimination was clearly done between EVD and other illness. Otherwise, authors can lay more on that.

2) I would like to read more about possible combination of the score with Ebola rapid diagnostic tests to see whether the performance could be increased.

3) What will be the authors’ advice on pauci-symptomatic cases if the score has to be used? What comment can drawn from this?

4) Lines 285-294: from the interpretation of the score, I do understand that any score presented will call for at least a 21 days follow-up. In this condition, I’m wondering if the score really helps discriminating EVD from other illnesses as stated in the manuscript?

5) Lines 295-305: I do not see in the text what the risk categorization brings in practices regarding case management at the triage unit.

6) The algorithm should reinforced with a decisional tree clarifying EVD suspects referral process or decision taken for care management.

7) Lines 315-319: can authors take the responsibility for discharging EVD suspects categorized as low or medium risk without two successive RT-PCR results within 48-72 hours? If yes, this should be stated in the manuscript. If not, authors should correct lines 311-319.

8) Can authors clearly assess the easiness of the tool to be used in the field conditions by health workers without mistaking or being overloaded, especially if more suspect-cases are received per day? Evaluating the time taken for consultation and experimentation of the score will be a key point to really know if this tool will be time saving or cumbersome for healthcare workers in the field.

9) Lines 330-331: I don’t understand what is written there.

10) Can authors say if this tool can be equally used at the tail and/or at the beginning of an EVD outbreak? Can lay a little bit more one that in the discussion section.

Conclusion

1) The conclusion varies from the study objective, can authors align this? The objective is stuck on the score and the conclusion focuses on RDTs…Not consistent…

References

References are inconsistently written. Correct.

6. PLOS authors have the option to publish the peer review history of their article (what does this mean?). If published, this will include your full peer review and any attached files.

**Do you want your identity to be public for this peer review?** For information about this choice, including consent withdrawal, please see our Privacy Policy.

Reviewer #1: No

Reviewer #2: No

While revising your submission, please upload your figure files to the Preflight Analysis and Conversion Engine (PACE) digital diagnostic tool, https://pacev2.apexcovantage.com/. PACE helps ensure that figures meet PLOS requirements. To use PACE, you must first register as a user. Registration is free. Then, login and navigate to the U

---

## [Editor Report · Decision Letter 1]

18 Jul 2024

Development of a clinical prediction score for Ebola virus disease screening at triage centers in the Democratic Republic of the Congo

PGPH-D-24-00337R1

Dear Dr Yango wa yango,

We are pleased to inform you that your manuscript 'Development of a clinical prediction score for Ebola virus disease screening at triage centers in the Democratic Republic of the Congo' has been provisionally accepted for publication in PLOS Global Public Health. You have addressed reviewers concerns thoroughly and the manuscript quality has been very much improved.

Best regards,

Maria del Pilar Fernandez

Academic Editor